# Uncertainty-aware Mean Teacher Framework with Inception and Squeeze-and-Excitation Block for MICCAI FLARE22 Challenge

Hui Meng[1], Haochen Zhao[2], Ziniu Yu[2], Qingfeng Li[2], and Jianwei Niu[2,3]

[1] School of Intelligent Science and Technology, Hangzhou Institute for Advanced Study, University of Chinese Academy of Sciences, 1 Sub-lane Xiangshan, Hangzhou 310024, China.
[2] Research center of Big Data and Computational Intelligence, Hangzhou Innovation Institute of Beihang University, Hangzhou 310051, China.
[3] State Key Laboratory of Virtual Reality Technology and Systems, School of Computer Science and Engineering, also with the Beijing Advanced Innovation Center for Big Data and Brain Computing (BDBC), Beihang University, Beijing 100191, China.

**Abstract.** Semi-supervised learning has attracted extensive attention in the field of medical image analysis. However, as a fundamental task, semi-supervised segmentation has not been investigated sufficiently in the field of multi-organ segmentation from abdominal CT. Therefore, we propose a novel uncertainty-aware mean teacher framework with inception and squeeze-and-excitation block (UMT-ISE). Specifically, the UMT-ISE consists of a teacher model and a student model, in which the student model learns from the teacher model by minimizing segmentation loss and consistency loss. Additionally, we adopt an uncertainty-aware algorithm to make the student model learn accurate and reliable targets by making full use of uncertainty information. To capture multi-scale features, the inception and squeeze-and-excitation block are incoporated into the UMT-ISE. It is worth noting that abdominal CT of test cases are first extracted before multi-organ segmentation in the inference phase, which significantly improves segmentation accuracy.We implement experiments on the FLARE22 challenge. Our method achieves mean DSC of 0.7465 on 13 abdominal organ segmentation tasks.

**Keywords:** semi-supervised learning, multi-organ segmentation, uncertainty estimation, multi-scale features

## 1 Introduction

Accurate segmentation of medical images is essential for many clinical applications, such as disease diagnosis and tumor localization [4]. Nowadays, manual segmentaion results given by radiologists are widely regarded as gold standards. However, manual segmentation is tedious and time consuming. Additionally, manual segmentation heavily depends on radiologists' experience and suffers

from intra- and inter-observer variabilities. Therefore, many researchers have developed different automatic segmentation methods [16], which are supposed to assist radiologists to make accurate diagnosis.

For abdominal organ segmentation, most research work focus on single organ segmentation, such as kidney [6] or blood vessels [9]. Compared with single-organ segmentation, multi-organ segmentation faces two major challenges. The first one is that large morphological differences between multiple organs limit accurate segmentation of all organs. The second one is that it's difficult to obtain large dataset with accurate annotations for multi-organ segmentation. Therefore, it is necessary to make full use of unlabeled medical images to improve the multi-organ segmentation accuracy [5].

To utilize unlabeled medical images effectively, we propose a novel UMT-ISE for segmenting multiple organs from 3D abdominal CT. The UMT-ISE is constructed based on conventional teacher-student model [3], which consists of a teacher model and a student model. For the same unlabeled data under different perturbations, the segmentation predictions of the teacher model and the student model are constrained to be consistent [17]. Different from the conventional teacher-student model, the UMT-ISE adopts framework of uncertainty-aware mean teacher (UA-MT) [17]. The teacher model in the UMT-ISE generates multiple predictions for each target under Monte Carlo sampling and gives uncertainty evaluation. The predictions with high uncertainty are filtered out and the predictions with low uncertainty are retained to compute consistency loss. Based on the design of the uncertainty evaluation, the teacher model tends to generate high-quality predictions and the student model can be constantly optimized. Considering multiple organs have different sizes, the inception and squeeze-and-excitation (ISE) block are incoporated into the UMT-ISE to capture multi-scale features.

## 2   Method

### 2.1   Preprocessing

The preprocessing operations can be divided into coarse segmentation and conventional data processing. Noted that the coarse segmentation is achieved by cropping CT-scans in z-axis, x-axis and y-axis directions, respectively. The detailed information of preprocessing operations are listed as follows:

– Cropping strategy in z-axis direction:
  The range of CT scans varies depending on the situation. For example, some patients may have CT scans not only of abdominal area, but of entire chest, lower abdomen and even legs. In some cases, only the abdominal region containing target organs is presented. Therefore, it is necessary to filter out some irrelevant and redundant slices in CT scans. In this study, we train an uncertainty-aware mean teacher (UA-MT) network to perform coarse abdominal segmentation.

To extract CT scans only containing abdominal region, we implement different cropping strategies in z direction. For training data with labels, we tailor them according to the range of target organs in annotations. For training data without labels, validation data and test data, we first implement coarse segmentation of target organs based on the trained UA-MT network and then crop CT scans according to the scope of segmentated organs.

During reference, we adopt specific preprocessing for large samples containing more than 800 slices and with z-axis spacing of 1. For these samples, we first equally divide the whole CT scans into three parts. Then, the coarse segmentation of target organs is implemented for each part. Finally, the CT scans containing target organs are extracted based on the segmentation results.

– Cropping strategy in x-axis and y-axis directions:
  According to observations, different samples have different proportions of target region to CT images in x-axis and y-axis directions. In some samples, the target organs only occupy a small region in CT images. It is necessary to cropping redundant background in x-axis and y-axis direction to enlarge the target organs. Conversely, the target organs in some samples occupy a large region in CT images. The target organs in these samples are close to the edge of the CT images, which resulting in mis-segmentation of target organs. For these samples, we pad the CT images with zero in x-axis and y-axis directions to ensure appropriate proportions of the target region to the corresponding CT images.
– Adjusting window level and window width:
  In order to achieve high contrast between the target organs and the background area, we adjust window width and window level of the original CT images. According to doctor's experience, the window width and window level of the CT images are adjusted to 40 and 255, respectively.
– Image Resampling:
  In this study, the network input of UMT-ISE is randomly cropped patches from whole CT images. The input size of the UMT-ISE is $112 \times 112 \times 80$, while the size of the whole CT images is much larger than $112 \times 112 \times 80$. To ensure the cropped patches contain efficient information, we resample all CT images to $192 \times 192 \times 96$ after the above preprocessing.
– Image normalization:
  After the above preprocessing, we implement z-score normalization on CT images based on the mean and standard deviation of the intensity values.
– Data augmentation:
  In this study, we implement random cropping on the whole CT images to obtain network input. Additionally, horizontal flipping are performed to achieve data augmentation.

### 2.2  Proposed Method

Strategies to use the unlabelled cases:

The input of teacher model and student model are the same CT images with different noises, and the output of the two models are constrained by unsupervised loss function.

Network architecture details:

The network architecture of the UMT-ISE is shown in Fig.1. The UMT-ISE is composed of two modified V-Net models, i.e. the teacher model and the student model, and the two models share the network structure. We update the teacher's weights as an exponential moving average (EMA) [17] of the student's weights to ensemble the information in different training step. Referring to the UA-MT [17], we estimate the uncertainty with the Monte Carlo Dropout[17]. In multi-organ segmentation, different organs have different size and the segmentation accuracy of small organs is low. In order to improve the accuracy of multi-organ segmentation, we insert ISE blocks in V-Net to obtain the modified V-Net. The ISE block mainly contains inception block and squeeze-and-excitation (SE) block, which can obtain multi-scale feature maps with channel attention. The network structure of the modified V-Net is shown in Fig.2.

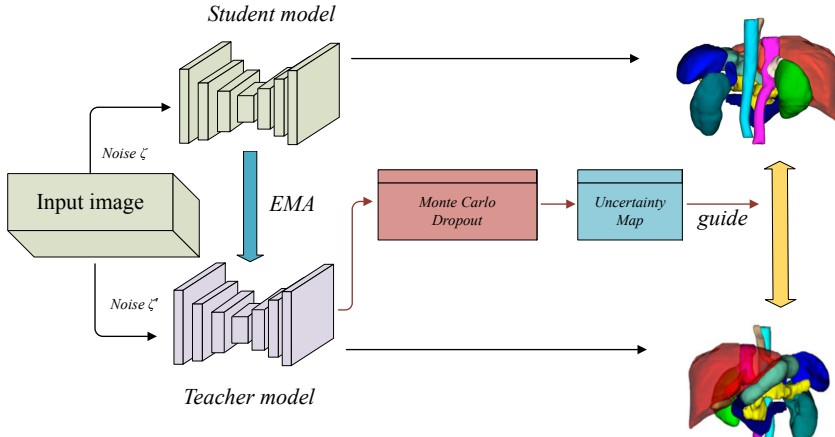

**Fig. 1.** Network architecture of the UMT-ISE. The network is constructed of a student model, a teacher model, and an uncertainty estimation module. The backbone of the student model and the teacher model is V-Net equipped with ISE blocks. The estimated uncertainty from the teacher model guides the student model to learn from the more reliable targets.

The structure of the ISE block is shown in Fig.3. The ISE block integrates the residual block, Inception block, and a SE block. Multiple convolution layers with different convolution kernels are used in the Inception block to obtain multiple feature maps with different receptive fields. Then, the feature maps are fused to generate multi-scale features to alleviate the impact of size diversity in multi-organ segmentation. The structure of the Inception block is shown in Fig.4(a).

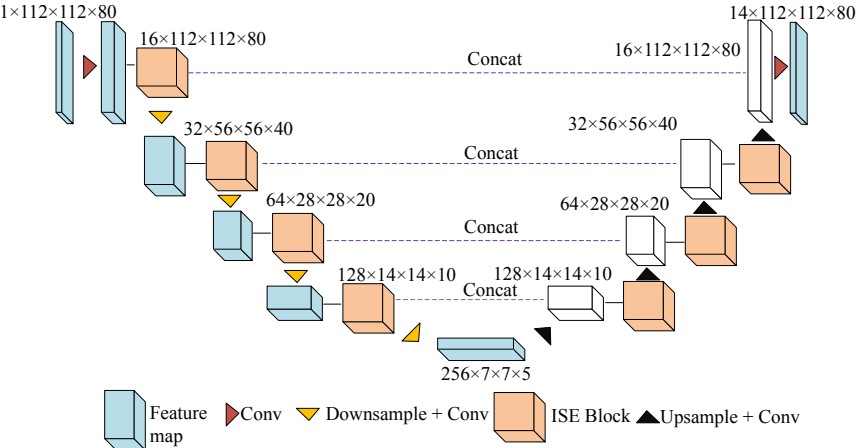

**Fig. 2.** Network architecture of the modified V-Net. The network is constructed of an encoder and a decoder, where four ISE blocks are inserted at the encoder and three ISE blocks are inserted at the decoder.

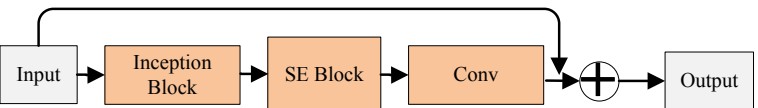

**Fig. 3.** The detailed architecture of the ISE block.

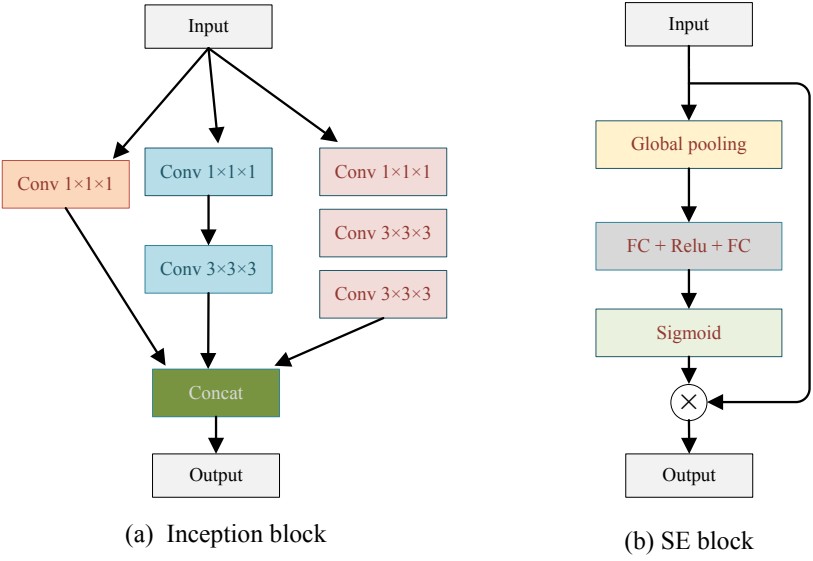

(a) Inception block

(b) SE block

**Fig. 4.** The detailed architecture of the Inception block (a) and the SE block (b).

Although the Inception block can enhance features of targets with different size, the redundant features in the multi-scale feature map reduce the discriminability of the network. Thus, a SE block is adopted to recalibrate the importance of the multi-scale features obtained by the Inception block. The structure of the SE block is illustrated in Fig.4(b). In the SE block, a global average pooling layer is used to aggregate the global information, which is followed by two fully connected (FC) layers to capture the channel-wise relationships. Then, the features given by the Inception block is recalibrated by the channel-wise relationships through point-wise multiplication.

Loss function:

In this study, we adopt Dice loss and cross entropy loss to calculate supervised loss on labeled data. Additionally, consistency loss on both unlabelled data and labeled data are calculated to optimize the network.

### 2.3    Post-processing

The post-processing operation used in this study is removing small connected areas to reduce false positive islands. Specifically, the largest connected area of each segmented organ is retained, and the other connected areas are removed.

## 3      Experiments

### 3.1    Dataset and evaluation measures

The FLARE 2022 is an extension of the FLARE 2021 [10] with more segmentation targets and more diverse images. The dataset is curated from more than 20 medical groups under the license permission, including MSD [14], KiTS [7,8], AbdomenCT-1K [11], and TCIA [2]. The training set includes 50 labeled CT scans with pancreas disease and 2000 unlabelled CT scans with liver, kidney, spleen, or pancreas diseases. The validation set includes 50 CT scans with liver, kidney, spleen, or pancreas diseases. The testing set includes 200 CT scans where 100 cases has liver, kidney, spleen, or pancreas diseases and the other 100 cases has uterine corpus endometrial, urothelial bladder, stomach, sarcomas, or ovarian diseases. All the CT scans only have image information and the center information is not available. The segmentation targets include 13 organs: liver, right kidney (RK), spleen, pancreas, aorta, inferior vena cava (IVC), right adrenal gland (RAG), left adrenal gland(LAG), gallbladder, esophagus, stomach, duodenum, and left kidney (LK).

The evaluation measures consist of two accuracy measures: Dice Similarity Coefficient (DSC) and Normalized Surface Dice (NSD), and three running efficiency measures: running time, area under GPU memory-time curve, and area under CPU utilization-time curve.

### 3.2    Implementation details

**Environment settings** The environments and requirements are presented in Table 1.

**Table 1.** Environments and requirements.

| | |
|---|---|
| Windows/Ubuntu version | Windows 10 |
| CPU | Intel(R) Core(TM) i9-9900K CPU @ 3.60GHz |
| RAM | 16×2GB; |
| GPU (number and type) | 1 NVIDIA Tesla V100 GPU (48G) |
| CUDA version | 11.1 |
| Programming language | Python 3.6 |
| Deep learning framework | Pytorch (Torch 1.7.0, torchvision 0.8.0) |
| Specification of dependencies None | |
| (Optional) Link to code | |

**Training protocols** In the training process, the batch size is set as 16, and the patch size is fixed as $80 \times 112 \times 112$. For optimization, we train our network for 2000 epochs using stochastic gradient descent (SGD) algorithm with an initial learning rate of 0.05 and a momentum of 0.9. During training, we use the poly scheduling to decay the learning rate. The calculation of the learning rate is as follows:

$$lr = lr_{base} \times (1 - \frac{epoch}{total\_epochs})^{0.9}$$

where $lr$ denotes the learning rate, and $lr_{base}$ is the initial learning rate. $epoch$ and $total\_epochs$ are current training epoch and the total training epochs, respectively.

Referring to the UA-MT [17], we use dice loss and the cross-entropy loss to calculate supervised loss. The consistency loss is adopted to calculate unsupervised loss. The total loss of our method is the weighted sum of the supervised loss and the unsupervised loss, which is defined as follows:

$$Loss_{total} = Loss_{CE} + Loss_{Dice} + \lambda *Loss_{consist}$$

where $Loss_{total}$ denotes total loss. $Loss_{CE}$, $Loss_{Dice}$ and $Loss_{consist}$ are the cross-entropy loss, Dice loss, and the consistency loss, respectively. $\lambda$ is an ramp-up weighting coefficient that controls the trade-off between the supervised and unsupervised loss.

## 4   Results

In this section, we assess the performance of the UMT-ISE using FLARE22 dataset. This section is arranged as follows: First, an ablation study for attention modules is implemented to verify the effectiveness of the ISE block. Second, we conduct ablation studies for improvement strategies including utilizing unlabelled data, coarse segmentation and post processing. Third, we evaluate our method on different backbones by replacing our backbone with V-Net [12], 3D U-Net [1], and Attention U-Net [13]. Fourth, we compare our method with baselines including mean teacher (MT) model [15] and the UA-MT model. Last,

**Table 2.** Training protocols.

| Network initialization | "he" normal initialization |
|---|---|
| Batch size | 16 |
| Patch size | 80×112×112 |
| Total epochs | 2000 |
| Optimizer | SGD with nesterov momentum ($\mu = 0.99$) |
| Initial learning rate (lr) | 0.05 |
| Lr decay schedule | lr = Initial learning rate $\times(1 - \frac{epoch}{total\_epochs})^{0.9}$ |
| Training time | 48 hours |
| loss function | cross entropy loss + Dice loss+$\lambda$ *consistency loss |
| Number of model parameters 9.44M[4] | |
| Number of flops | 41.40G[5] |
| $CO_2$eq | 1 Kg[6] |

the segmentation results on test set and qualitive results of our method are presented.

### 4.1    Ablation study for attention modules

The ablation study is implemented to evaluate the effectiveness of the ISE block. The baseline of the ablation study is the UA-MT network with our proposed preprocessing and post-processing operations. Table 3 lists quantitative results of different networks on validation set. Compared with the baseline, employing the SE block individually yields a result of 0.7259 in mean DSC, which represents 4.33% improvement. Additionally, the network with only the inception block achieves 0.7446 in mean DSC, which outperforms the baseline by 6.2%. Furthermore, integration of the inception block and the SE block (i.e., UMT-ISE) yields the highest mean DSC (0.7465). These comparisons illustrate that the inception block and the SE block have potential to improve the accuracy of multi-organ segmentation.

### 4.2    Ablation study for improvement strategies

To validate the superiority of our method in utilizing unlabelled data, we trained two models based on our method using different data. The first one is trained with only labeled data, and the second one is trained with both labeled and unlabelled data. The two models are tested on validation set, and the DSCs given by the two models are listed in Table 4. Compared with the first model, the DSCs of most organs given by the second model are higher. Additionally, the mean DSC given by the second model is 0.7465, which outperforms that of

**Table 3.** Quantitative results of ablation experiments for attention modules.

| Attention module | None | SE block | Inception block | ISE block |
|---|---|---|---|---|
| Liver | 0.9260 | 0.9338 | 0.9527 | 0.9549 |
| RK | 0.8278 | 0.8357 | 0.8493 | 0.8499 |
| Spleen | 0.8588 | 0.8616 | 0.8650 | 0.8890 |
| Pancreas | 0.6493 | 0.6944 | 0.7099 | 0.7155 |
| Aorta | 0.8467 | 0.8377 | 0.8367 | 0.8490 |
| IVC | 0.6499 | 0.7444 | 0.7738 | 0.7687 |
| RAG | 0.5107 | 0.5832 | 0.5723 | 0.6115 |
| LAG | 0.5076 | 0.4919 | 0.5716 | 0.5168 |
| Gallbladder | 0.5916 | 0.6297 | 0.6475 | 0.6431 |
| Esophagus | 0.5760 | 0.6663 | 0.6712 | 0.6822 |
| Stomach | 0.6065 | 0.7855 | 0.7911 | 0.8160 |
| Duodenum | 0.4770 | 0.5322 | 0.6274 | 0.5472 |
| LK | 0.8454 | 0.8408 | 0.8115 | 0.8602 |
| Mean DSC | 0.6826 | 0.7259 | 0.7446 | 0.7465 |

**Table 4.** Comparison results of our models trained with and without unlabelled data.

| Training data | Only labeled data | labeled and unlabeled data |
|---|---|---|
| Liver | 0.9221 | 0.9549 |
| RK | 0.8280 | 0.8499 |
| Spleen | 0.8118 | 0.8890 |
| Pancreas | 0.7148 | 0.7155 |
| Aorta | 0.8020 | 0.8490 |
| IVC | 0.7331 | 0.7687 |
| RAG | 0.6158 | 0.6115 |
| LAG | 0.5473 | 0.5168 |
| Gallbladder | 0.5677 | 0.6431 |
| Esophagus | 0.6548 | 0.6822 |
| Stomach | 0.7629 | 0.8160 |
| Duodenum | 0.5748 | 0.5472 |
| LK | 0.7465 | 0.8602 |
| Mean DSC | 0.7193 | 0.7465 |

the first model by 2.72%. These results demonstrate that the utilization of the unlabelled data can improve the segmentation performance in our method.

To evaluate the effectiveness of the coarse segmentation and the post processing, we implemented inference experiments on validation set with different strategies. The baseline is our model tested without coarse segmentation and post processing. Table 5 lists quantitative results of our model tested with different inference strategies. Compared with the baseline, conducting the coarse segmentation individually yields a significantly higher result of 0.7461 in mean DSC, which represents 12.85% improvement. Additionally, inference with only the post processing obtains higher mean DSC (0.6275) than that of the baseline (0.6176). Furthermore, implementing both the coarse segmentation and the post processing achieves the highest mean DSC (0.7465). All these comparisons demonstrate that the coarse segmentation and the post processing can effectively improve accuracy of multi-organ segmentation.

**Table 5.** Quantitative results of ablation study for coarse segmentation and post processing.

| Inference strategy | None | only coarse segmentation | only post processing | coarse segmentation and post processing |
|---|---|---|---|---|
| Liver | 0.9116 | 0.9514 | 0.9136 | 0.9549 |
| RK | 0.7408 | 0.8370 | 0.7942 | 0.8499 |
| Spleen | 0.8218 | 0.8841 | 0.7911 | 0.8890 |
| Pancreas | 0.5801 | 0.7150 | 0.6356 | 0.7155 |
| Aorta | 0.7592 | 0.8459 | 0.7056 | 0.8490 |
| IVC | 0.6967 | 0.7809 | 0.6849 | 0.7687 |
| RAG | 0.5014 | 0.6107 | 0.3799 | 0.6115 |
| LAG | 0.2595 | 0.5180 | 0.3969 | 0.5168 |
| Gallbladder | 0.4275 | 0.6296 | 0.6002 | 0.6431 |
| Esophagus | 0.5803 | 0.7029 | 0.5457 | 0.6822 |
| Stomach | 0.6081 | 0.8106 | 0.5802 | 0.8160 |
| Duodenum | 0.3814 | 0.5540 | 0.4581 | 0.5472 |
| LK | 0.7606 | 0.8595 | 0.6711 | 0.8602 |
| Mean DSC | 0.6176 | 0.7461 | 0.6275 | 0.7465 |

### 4.3    Experiments on different backbones

To evaluate the performance of the UMT-ISE over different backbones, we replaced backbones of the teacher model and the student model with V-Net [12], 3D U-Net [1], and attention U-Net [13], respectively. The proposed preprocessing and post-processing operations are conducted for all models in comparison

experiments. Table 6 lists quantitative results of different networks on validation set. Compared with the network with the V-Net as the backbone, our method achieves significantly higher mean DSC. Additionally, the 3D U-Net and the attention U-Net obtain higher mean DSC than the V-Net. Furthermore, our method achieves the highest mean DSC (0.7465). These comparisons further indicate the efficiency of the UMT-ISE in multi-organ segmentation.

**Table 6.** Quantitative results of networks with different backbones on validation set.

| Backbone | V-Net | 3D U-Net | Attention U-Net | V-Net+ISE block |
|---|---|---|---|---|
| Liver | 0.9260 | 0.9542 | 0.9487 | 0.9549 |
| RK | 0.8278 | 0.8521 | 0.8269 | 0.8499 |
| Spleen | 0.8588 | 0.9106 | 0.8686 | 0.8890 |
| Pancreas | 0.6493 | 0.6523 | 0.7070 | 0.7155 |
| Aorta | 0.8467 | 0.8574 | 0.8532 | 0.8490 |
| IVC | 0.6499 | 0.6592 | 0.7090 | 0.7687 |
| RAG | 0.5107 | 0.4695 | 0.5710 | 0.6115 |
| LAG | 0.5076 | 0.5499 | 0.5961 | 0.5168 |
| Gallbladder | 0.5916 | 0.6703 | 0.6236 | 0.6431 |
| Esophagus | 0.5760 | 0.6538 | 0.6498 | 0.6822 |
| Stomach | 0.6065 | 0.7243 | 0.7180 | 0.8160 |
| Duodenum | 0.4770 | 0.4339 | 0.4648 | 0.5472 |
| LK | 0.8454 | 0.8659 | 0.8511 | 0.8602 |
| Mean DSC | 0.6826 | 0.7118 | 0.7222 | 0.7465 |

### 4.4  Comparison experiments with baselines

The UMT-ISE is constructed based on the UA-MT model, which is generated by modifying the MT model [15]. To validate the superiority of the UMT-ISE over the UA-MT and the MT model, we trained and tested the conventional UA-MT and MT model using FLARE22 dataset. Tabel 7 lists quantitative results of different methods on validation set. Compared with the MT model, the UA-MT model obtains higher mean DSC (0.5905). Additionally, the UMT-ISE achieves the highest mean DSC (0.7465), which outperforms the MT and the UA-MT by 16.21% and 15.60%, respectively. These results verify the effectiveness of the uncertainty-aware scheme, the ISE block, the coarse segmentation and the post processing in our method.

### 4.5  Segmentation results of our method

Table 8 lists quantitative results of our method on testing set. The mean DSC and NSD are 0.7104 and 0.7763, respectively. Consistent with the validation

**Table 7.** Quantitative results of different methods on validation set.

| Method | MT | UA-MT | UMT-ISE |
|---|---|---|---|
| Liver | 0.8834 | 0.8982 | 0.9549 |
| RK | 0.7110 | 0.7319 | 0.8499 |
| Spleen | 0.7388 | 0.7754 | 0.8890 |
| Pancreas | 0.5254 | 0.4783 | 0.7155 |
| Aorta | 0.7849 | 0.7960 | 0.8490 |
| IVC | 0.5868 | 0.6487 | 0.7687 |
| RAG | 0.4268 | 0.3712 | 0.6115 |
| LAG | 0.3638 | 0.2846 | 0.5168 |
| Gallbladder | 0.4996 | 0.4641 | 0.6431 |
| Esophagus | 0.5322 | 0.5835 | 0.6822 |
| Stomach | 0.4612 | 0.5173 | 0.8160 |
| Duodenum | 0.3741 | 0.3935 | 0.5472 |
| LK | 0.7095 | 0.7336 | 0.8602 |
| Mean DSC | 0.5844 | 0.5905 | 0.7465 |

results, the segmentation of liver achieves the highest DSC (0.9501) and the segmentation of LAG obtains the lowest DSC (0.5201).

**Table 8.** Quantitative results of our method on testing set.

| Organ | DSC | NSD |
|---|---|---|
| Liver | 0.9501 | 0.9495 |
| RK | 0.8444 | 0.8647 |
| Spleen | 0.8243 | 0.8364 |
| Pancreas | 0.6535 | 0.7732 |
| Aorta | 0.7988 | 0.8411 |
| IVC | 0.7132 | 0.7191 |
| RAG | 0.6036 | 0.7818 |
| LAG | 0.5201 | 0.6593 |
| Gallbladder | 0.6094 | 0.5911 |
| Esophagus | 0.6273 | 0.7472 |
| Stomach | 0.7885 | 0.8038 |
| Duodenum | 0.4674 | 0.6675 |
| LK | 0.8348 | 0.8564 |
| Mean | 0.7104 | 0.7763 |

Fig.5 and Fig.6 show examples with good segmentation results and bad segmentation results, respectively. As for the bad segmentation cases, we think there are three reasons. The first one is that the low imaging quality of the CT images causes the bad segmentation. Specifically, there are dark holes in some

organs which results in broken segmentation results of the organs. As shown in Case#0048 and Case#0042 (Fig.6), the segmentation results of stomach are incomplete because of interference of dark holes. The second one is that our model is not robust enough for accurate segmentation of small organs. As shown in Case#0048 (Fig.6), the left kidney is not segmented. The last one is that the target organs only occupy a small region in some CT images (Case#0028), which increases the difficulty of segmentation. Although the coarse segmentation can enlarge abdominal region in CT images, the image quality is decreased.

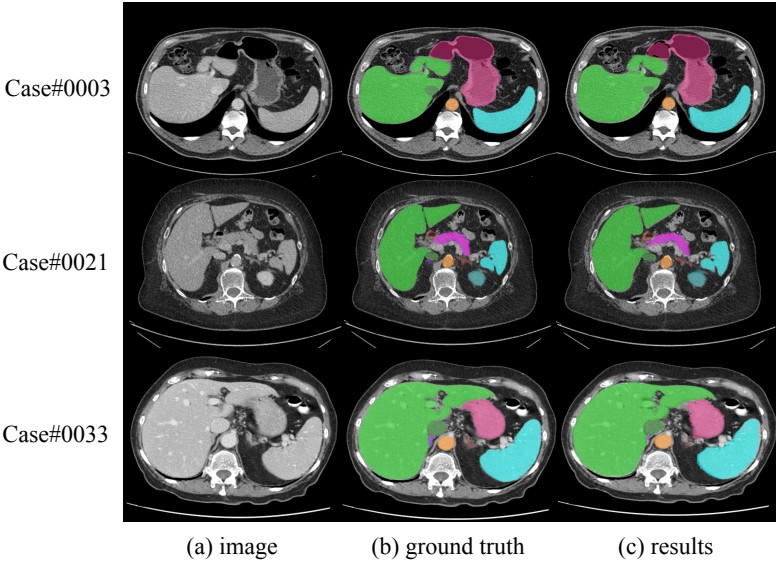

**Fig. 5.** Well-segmented examples from validation sets.

To optimize inference efficiency of our method, we adopt coarse segmentation to crop redundant slices, which reduces calculation of the UMT-ISE during inference. Additionally, we use CT scans resampled to $96 \times 192 \times 192$ during inference rather than using original CT scans. Furthermore, patch-based segmentation is implemented in inference which optimizes inference efficiency. To evaluate the inference efficiency of our method, we run our trained model on a docker with NVIDIA 2080Ti GPU(12GB) and 32GB RAM for the 50 validation cases. The average inference time is 56.11 seconds, and the maximum GPU memory used is 2.98GB. Noted that validation Case 10 and 50 are scans of full body, which consume 103.31 and 237.89 seconds, respectively. The average area under GPU memory-time curve and area under CPU utilization-time curve are 152226 and 951, respectively.

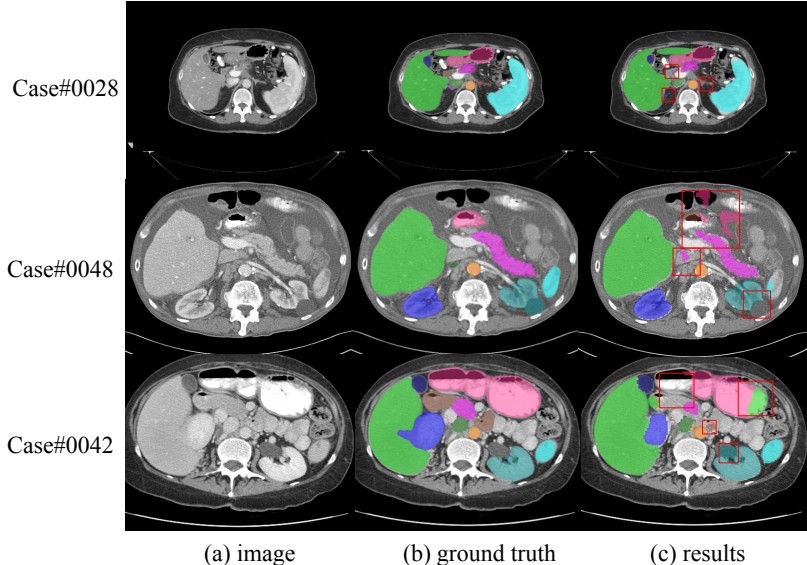

Case#0028

Case#0048

Case#0042

(a) image          (b) ground truth          (c) results

**Fig. 6.** Challenging examples from validation sets.

## 5   Conclusion

In this study, we propose a novel UMT-ISE for multi-organ segmentation in abdominal CT. The UMT-ISE achieves fast and accurate multi-organ segmentation. Additionally, our method can be tested on CPU, which is convenient to complete some clinical tasks. However, our method still has some limitations. For some small organs, their shapes and positions are easily affected by tumors and edema. Our method is not robust enough for segmentation of small organs. Additionally, it is difficult to extract abdominal regions for the cases with many CT slices, and the segmentation results of these cases are not satisfied. Furthermore, the coarse segmentation improves the final segmentation accuracy, but increases inference time to some extent. Our future work will focus on the accurate segmentation of small organs in multi-organ segmentation and develop more fast and accurate segmentation methods.

## 6   Acknowledgment

The authors of this paper declare that the segmentation method they implemented for participation in the FLARE22 challenge didn't use any pre-trained models or additional datasets other than those provided by the organizers.

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
