# OpenReview forum: "Uncertainty-aware Mean Teacher Framework with Inception and Squeeze-and-Excitation Block for MICCAI FLARE22 Challenge"
_MICCAI.org/2022/Challenge/FLARE_

### Official Review · Reviewer_1qMn · 2022-09-16
**Good framework for the unlabeled data**

**Rating:** 7
**Confidence:** 3

**Review:**

Pros:
1.  A novel uncertainty-aware mean teacher framework with inception and squeeze-and-excitation block (UMT-ISE) is proposed, and which could enhance the learning accurate and reliable targets.

Cons:
1. More details of the novel framework would be useful for understanding this work, e.g., Monte Carlo Dropout, Uncertainty Map, EMA, etc.

---

> ### Author Response · Authors · 2022-10-14
> **More details of our framework are added**
>
> Thanks for your professional suggestion. All changes in revised manuscript are labeled with red font.
> - We have added more details of our proposed method in Section 2.2 and Section 3.2.

---

### Official Review · Reviewer_N79B · 2022-09-16
**Uncertainty-aware mean teacher framework with consistency regularization for semi-supervised organ segmentation**

**Rating:** 6
**Confidence:** 4

**Review:**

The authors propose an uncertainty-aware mean teacher framework with consistency regularization for semi-supervised organ segmentation.

Pros:
. Well-written paper, methods are clearly described.
. The method achieves relatively good performance on the validation set with a mean Dice of 0.7458


Cons:
. No results are presented in the abstract.
. No ablation study is performed to evaluate the individual contributions of all introduced components, namely the uncertainty-based selection of unlabeled samples and the post-processing.
. The performance using the unlabeled data improves by less than 1% compared to the fully-supervised baseline. This indicates that the proposed semi-supervised method cannot efficiently leverage the unlabelled data.

---

> ### Author Response · Authors · 2022-10-14
> **Ablation studies have been implemented**
>
> Thanks for your professional suggestions. All changes in revised manuscript are labeled with red font.
> - We have added quantitative results of our method in the abstract.
> - To evaluate the effectiveness of the uncertainty-based selection of unlabelled samples, we implemented experiments based on conventional mean teacher (MT) model and uncertainty-aware mean teacher (UA-MT) model. The quantitative results are listed in Table 7, and the analysis is presented in Section 4.4.
> - We implemented ablation study for improvement strategies including coarse segmentation and post processing. The quantitative results are listed in Table 5, and the analysis is presented in Section 4.2.
> - Table 4 lists comparison results of our models trained with and without unlabelled data. Compared with the model trained with only labeled data, the model trained with unlabelled data achieves a result of 0.7465 in mean DSC, which represents 2.7% improvement. These results demonstrate that our method can efficiently leverage the unlabelled data.

---

### Official Review · Reviewer_D7KW · 2022-09-19
**A good try, but lack of innovations**

**Rating:** 5
**Confidence:** 5

**Review:**

Pros:
Preprocessing phase was well described
Mean teacher framework based on VNet was used for this semi-supervised segmentation task. A inception block and a squeeze-and-excitation block was added into the VNet.

Cons:
Mean teacher framework is a well-known method of semi-supervised learning, while the inception block and squeeze-and-excitation block are also common method.
Why combining these two can theoretically improve the multi-organ segmentation performance was not discussed.
Overall, the article is lack of innovations, the modifications with mean teacher framework are trivial.

---

> ### Author Response · Authors · 2022-10-14
> **Effectiveness of attention modules is verified**
>
> We feel great thanks for your professional review on our article. All changes in revised manuscript are labeled with red font.
> - In multi-organ segmentation, especially abdominal organ segmentation, the volume of organs is extremely different. For example, liver occupies a large volume in abdominal CT, whereas left adrenal gland has a small volume. Additionally, we find that the segmentation results of small organs given by conventional mean teacher model are poor. We think it is because of the encoder-decoder structure of V-Net, small organs cannot retain sufficient information in deep feature maps. To improve the accuracy of multi-organ segmentation, we insert the inception block and the squeeze-and-excitation (SE) block in V-Net to enhance feature representation. The discussion is presented in Section 2.2.
> - To verify the effectiveness of the inception block and the SE block, we implemented ablation study for attention modules. The quantitative results are listed in Table 3. Compared with the V-Net, employing the SE block individually yields a result of 0.7259 in mean DSC, which represents 4.33% improvement. Additionally, the network with only the inception block achieves 0.7446 in mean DSC, which outperforms the baseline by 6.2%. Furthermore, integration of the inception block and the SE block (i.e., UMT-ISE) yields the highest mean DSC (0.7465). These comparisons illustrate that the inception block and the SE block have potential to improve the accuracy of multi-organ segmentation.
> - Compared with the conventional mean teacher framework, we add coarse segmentation, post-processing, ISE block, uncertainty-aware scheme in our method for FLARE22 Challenge. The results of ablation studies verify the effectiveness of the introduced components. The quantitative results and analysis are presented in Section 4.

---

### Official Review · Reviewer_1hcz · 2022-09-20
**Overall good paper**

**Rating:** 8
**Confidence:** 4

**Review:**

The description of the method is clear and easy to follow. The motivation of the method is convinsive.

Comments:
1. It would be interesting to compare the performance of your method with conventional teacher-student model to clearly see how filtering of high uncertainty predictions affect the quality of the segmentation and probably feature maps.
2. More details about the method would be helpful, for example what "unsupervised loss function" did you use?

---

> ### Author Response · Authors · 2022-10-14
> **Comparison experiments are implemented and more details are added**
>
> Thanks for your professional suggestions. All changes in revised manuscript are labeled with red font.
> 1. We have implemented experiments based on conventional teacher-student model (i.e., mean teacher, MT) and uncertainty-aware mean teacher (UA-MT) model. The quantitative results on validation set are listed in Table 7, and the analysis is presented in Section 4.4. As shown in Table 7, The UA-MT model achieves higher mean DSC (0.5905) than that of the MT model, which verifies the effectiveness of the uncertainty-aware scheme.
> 2. We have added more detailed description of our method. The loss description is added in Section 3.2.

---

### Official Review · Reviewer_rwLd · 2022-09-22
**Good work merging mean teacher training and SE Block with U-Net on multi-organ segmentation task**

**Rating:** 9
**Confidence:** 3

**Review:**

Pros:
- The network speed and metric scores are balanced
- By mean teacher training and Monte Carlo sampling, the performance is improved and the model learns from unlabelled data
- The preprocessing is detailed and considerate

Cons:
- It's better to describe the post processing and explain more about cases in discussion section

---

> ### Author Response · Authors · 2022-10-14
> **Description and explaination have been added**
>
> Thanks for your careful review on our article.
> - We have added description of the post processing in Section 2.3 and explain more about cases in Section 4.5. All changes in revised manuscript are labeled with red font.

---

### Meta-Review · Program_Chairs · 2022-09-28

**Recommendation:** Major Revision
**Confidence:** 5

**Metareview:**

Reviewers raise many concerns and suggestions. Please address all comments in the revised manuscript.

---

> ### Author Response · Authors · 2022-10-14
> **Revision Summary**
>
> Thanks for your review on our article. We have revised our previous manuscript based on the comments raised by the reviewers. All changes in revised manuscript are labeled with red font. The main revisions are summarized as follows:
> - We implemented ablation study for attention modules to verify the effectiveness of the ISE block. The quantitative results are listed in Table 3, and the statistical analysis is presented in Section 4.1.
> - We implemented ablation study for improvement strategies including coarse segmentation and post processing. The quantitative results are listed in Table 5, and the analysis is presented in Section 4.2.
> - To verify the effectiveness of our modified V-Net, we implemented experiments on different backbones. The quantitative results are listed in Table 6, and the analysis is presented in Section 4.3.
> - We implemented experiments based on conventional mean teacher model and uncertainty-aware mean teacher model. The quantitative results are listed in Table 7, and the analysis is presented in Section 4.4.